# DAC-DETR: Divide the Attention Layers and Conquer

**Zhengdong Hu**[1,2*], **Yifan Sun**[2], **Jingdong Wang**[2], **Yi Yang**[3] [†]
[1] ReLER, AAII, University of Technology Sydney
[2] Baidu Inc.
[3] CCAI, College of Computer Science and Technology, Zhejiang University
`huzhengdongcs@gmail.com,sunyf15@tsinghua.org.cn,`
`wangjingdong@baidu.com,yangyics@zju.edu.cn`

## Abstract

This paper reveals a characteristic of DEtection Transformer (DETR) that negatively impacts its training efficacy, *i.e.*, the cross-attention and self-attention layers in DETR decoder have opposing impacts on the object queries (though both impacts are important). Specifically, we observe the cross-attention tends to gather multiple queries around the same object, while the self-attention disperses these queries far away. To improve the training efficacy, we propose a Divide-And-Conquer DETR (DAC-DETR) that separates out the cross-attention to avoid these competing objectives. During training, DAC-DETR employs an auxiliary decoder that focuses on learning the cross-attention layers. The auxiliary decoder, while sharing all the other parameters, has NO self-attention layers and employs one-to-many label assignment to improve the gathering effect. Experiments show that DAC-DETR brings remarkable improvement over popular DETRs. For example, under the 12 epochs training scheme on MS-COCO, DAC-DETR improves Deformable DETR (ResNet-50) by +3.4AP and achieves 50.9 (ResNet-50) / 58.1 AP (Swin-Large) based on some popular methods (*i.e.*, DINO and an IoU-related loss). Our code will be made available at `https://github.com/huzhengdongcs/DAC-DETR`.

## 1   Introduction

Though DEtection Transformers (DETRs) have already achieved great progress, their low training efficacy problem remains a critical challenge. The research community has been paying great efforts to locate the causes of this problem and to improve the training efficacy. For example, DN-DETR [15] and DINO [42] find the one-to-one matching is prone to inconsistent optimization goals. In response, they propose to accelerate DETR training with auxiliary denoising queries. Hybrid DETR [12] and Group DETR [5] attribute the reason to the rare positive training queries and correspondingly increase positive queries through additional one-to-many matching. Some other works [20, 1] notice the misalignment between the predicted probability and position. They suppress such misalignment through IoU-related loss functions and achieve improvement.

This paper, from another perspective, investigates this problem by rethinking the cooperation between cross-attention and self-attention layers in the DETR decoder. We find during their cooperation, these two attention types actually have some opposing impacts on the object queries. These two impacts are both critical for DETR (as explained later), but their opposition impairs the training efficacy. Specifically, we observe the cross-attention layers in DETR decoder tends to gather multiple queries around a single object, while the self-attention disperses these queries far away from each other. The above "gather ↔ disperse" phenomenon concerns not only the positional relationship (1st row in Fig. 1), but also the feature distance (2nd row in Fig. 1 visualized by t-SNE [36]):

---

[*]Zhengdong Hu makes his part of work during internship in Baidu Inc.
[†]Corresponding author.

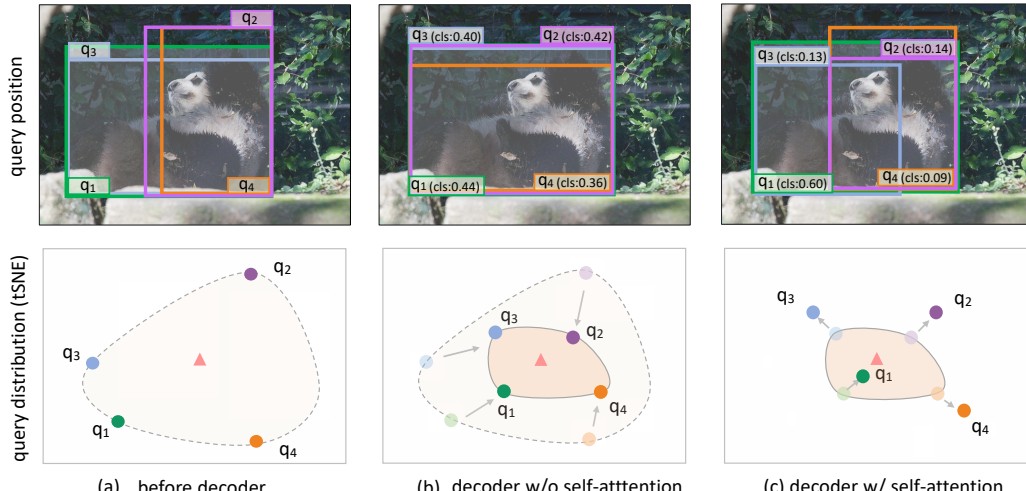

Figure 1: Comparison between (a) initial queries before the DETR decoder, (b) queries output from a decoder variant without self-attention layers and (c) queries output from the normal DETR decoder. We visualize their position (1st row) and feature distribution (2nd row). For clarity, we randomly choose only 4 queries with relatively large IoU. From (a) → (b), multiple queries are pulled towards a same object, showing the gathering effect of cross-attention layers. From (b) → (c), adding the self-attention layers disperses these queries except a single one ($q_1$), which approaches even closer towards their center. In spite of the conflicting objectives, we find that these two effects are both important for DETR.

• *Cross-attention layers tend to gather multiple queries around the same object.* Given an already-trained Deformable DETR [45], we remove all the self-attention layers in its decoder. This removal compromises DETR to duplicate detections. As illustrated in Fig. 1 (b), multiple queries locate a same object ("bear") with relatively large Intersection-over-Union (IoU). Compared with their initial states (before the decoder) in Fig. 1 (a), they become closer towards each other, regarding both their position (1st row) and feature distance (2nd row).

• *Self-attention layers disperse these queries from each other.* In Fig. 1 (c), we restore the original deformable DETR. Correspondingly, the queries are dispersed from each other regarding both position (1st row) and feature distance (2nd row). Due to this dispersion, most queries become farther to their original center point, except that a single query (with the highest classification score) approaches even closer. This phenomenon explains how DETR makes non-duplicate detection, and is consistent with the consensus [23, 3, 41, 25], *i.e.*, self-attention layers play a critical role in removing duplicates.

In spite of their opposition, these two effects are both critical for DETR. The gathering effect allows the queries to aggregate feature from the nearby object. The dispersing effect allows the self-attention layers to remove duplicate detection. However, learning the stacked self-attention and cross-attention layers within such "gather ↔ disperse" opposition is difficult. To improve the training efficacy, we propose a Divide-and-Conquer DETR (DAC-DETR) that *divides* the cross-attention out from this opposition for better *conquering*. We explain the key points of DAC-DETR in more details as below:

**1)** "*Divide*": DAC-DETR employs an auxiliary decoder that focuses on learning the cross-attention layers. The auxiliary decoder has NO self-attention layers and shares all the other parameters (*i.e.*, the cross-attention layers and the feed-forward networks) with the original decoder (O-Decoder). We name the auxiliary decoder as C-Decoder to highlight its focus on the cross-attention.

**2)** "*Conquer*": During training, DAC-DETR feeds all the queries into two decoders in parallel. O-Decoder uses the one-to-one matching to learn non-duplicate detection. In contrast, C-Decoder uses the one-to-many matching that assigns each object with multiple positive queries. C-Decoder has no self-attention layers and is prone to duplicate detection results. Therefore, using one-to-many label assignment is a natural choice and benefits learning the gathering effect with more positive queries. During inference, DAC-DETR discards C-Decoder and maintains the baseline efficiency.

Through the above "divide-and-conquer", DAC-DETR defuses the opposition when learning the "gathering" and "dispersing" effects which are both important for DETR. Mechanism analysis in Section 3.3 shows that due to the enhanced feature aggregation ability, DAC-DETR improves both the quantity and quality of intermediate queries that are gathered to each ground-truth object. Based on these improved queries, DETR increases the detection accuracy (after removing duplicate predictions).

We conduct extensive experiments to validate the effectiveness of DAC-DETR and empirically show remarkable improvement over various DETRs. For example, based on a popular baseline, *i.e.*, ResNet-50 Deformable DETR [45], DAC-DETR brings +3.4 AP improvement and achieves 47.1 AP on MS-COCO within 12 (1×) training epochs. On some more recent state-of-the-art methods (that usually integrate a battery of good practices), DAC-DETR still gains consistent and complementary benefits. For example, under the 1× learning scheme on MS-COCO, DAC-DETR improves ResNet-50 DINO to 50.0 (+1.0) AP. Combining an IoU-related Loss in Align DETR [1], our method further achieves 50.9 AP (ResNet-50) / 58.1 AP (Swin-Large), surpassing Align DETR itself by +0.7 AP. Moreover, the achieved results (50.9 and 58.1 AP) are also higher than Stable DETR [20] (50.4 and 57.7 AP), which adopts even more good practices (*i.e.*, stable matching and memory fusion).

Our main contributions are summarized as follows: First, we reveal the "gather ↔ disperse" opposition between cross-attention and self-attention layers in DETR decoder as a reason for its low training efficacy. Second, we propose to divide the cross-attention layers out from this opposition for better conquering the DETR training, yielding the so-called Divide-and-Conquer DETR (DAC-DETR). Third, we empirically show that DAC-DETR brings remarkable improvement over popular DETR baselines and performs favorably against most recent state-of-the-art methods.

## 2 Related works

**DETR-like object detectors.** The original Detection Transformer (DETR) [37, 3] is featured for its end-to-end detection without any hand-crafted modules (*e.g.*, Non-maximum Suppression). Since DETR [3] suffers from low training convergence, many efforts [45, 33, 23, 42, 38, 6, 8, 34, 43, 39, 32, 16] have been paid to improve the training efficacy. Deformable DETR [45] proposes to replace traditional attention modules with deformable attention and improves the training convergence. Conditional DETR [23] performs the conditional spatial query for decoder multi-head cross-attention. DAB-DETR [19] utilizes box coordinates as priors and dynamically updates anchor boxes layer-by-layer. DN-DETR [15] introduces denoising part to stabilize the bipartial matching, and DINO [42] further adds contrastive learning to enhance the performance. Some works [20, 1] propose IoU-related loss functions to align the predicted probability score against position. This paper, from another perspective, finds a opposing impact between the cross-attention and self-attention layers and proposes to divide the attention layers during training.

**Attention layers in DETR decoder.** The cross-attention and self-attention layers are both essential for DETRs. The former allows the object queries to aggregate information from the image features, while the latter facilitates query-to-query interaction to remove duplicate detection [23, 45, 3, 2, 9, 40, 31]. Our observation in this paper is consistent with the above consensus and goes a step forward: the cross-attention layers tend to gather multiple queries towards each object, while the self-attention disperses them far away. By separating the cross-attention layers out from this opposition, the proposed DAC-DETR improves DETR training efficacy.

**One-to-many label assignment.** One-to-many label assignment is widely utilized in CNN-based detectors [29, 27, 28, 11, 4, 35, 13]. these methods produce duplicate predictions on each object and uses hand-crafted non-maximum suppression (NMS) [24] for post-processing. As the DETRs prevail, some recent literature [5, 12, 26] explore one-to-many label assignment for DETRs and demonstrate benefit. Group DETR [5] employs multiple parallel object queries, which jointly contribute multiple positive queries for each object. Hybrid DETR [12] adds an auxiliary query branch that performs one-to-many label assignment.

DAC-DETR uses the one-to-many label assignment for the auxiliary C-Decoder, because C-Decoder has no self-attention layers that help removing duplicate predictions. Therefore, DAC-DETR also gains the benefit from one-to-many assignment. That being said, the benefit from one-to-many assignment is not our key contribution but a natural gain. While a co-current work Hybrid DETR [12] already shows that adding a one-to-many decoder improves DETR, our method shows that removing

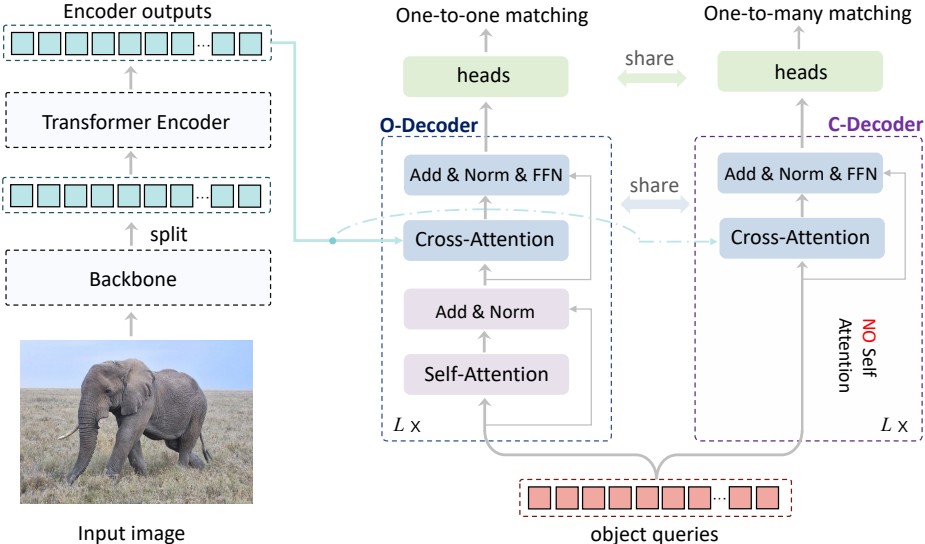

Figure 2: Overview of DAC-DETR. Left: for the backbone and encoder, DAC-DETR follows the standard DETR pipeline: it uses a backbone (*e.g.*, ResNet-50) to extract feature, splits the feature into patch tokens and then feeds the tokens into the transformer encoder. The tokens output from the encoder are fed into the decoders for object detection. Right: the main difference between DAC-DETR and previous DETRs is that DAC-DETR has two parallel decoders, *i.e.*, an O-Decoder (standard decoder) and an auxiliary C-Decoder (NO self-attention layers). These two decoders share most parameters and the same object queries, except that C-Decoder has no self-attention layers. O-Decoder is supervised with one-to-one matching to learn non-duplicate detection. In contrast, Decoder uses one-to-many matching to assign each ground-truth object with multiple positive queries. During inference, DAC-DETR only uses the O-Decoder and maintains the baseline efficiency.

the self-attention in the one-to-many decoder is better (e.g., +0.6 AP on MS-COCO in 12 training epochs, as detailed in Table 6 ).

## 3 Methods

### 3.1 Overview

**Preliminaries on DETRs.** A DETR-style detector typically consists of a backbone (*e.g.*, ResNet [10], Swin Transformer [21]), a transformer encoder and a transformer decoder. Given an input image, the DETR-style detector first feeds it into the backbone to extract image features, splits the features into patch tokens and then enhances the patch tokens by the transformer encoder. The enhanced tokens from the encoder are $\mathbf{Z} = \{z_1, ..., z_m\}$. The transformer decoder takes multiple object queries $\mathbf{Q} = \{q_1, ..., q_n\}$ as its input. The decoder has $L$ stacked decoder blocks. In each decoder block, the object queries $\mathbf{Q}$ sequentially undergo a self-attention layer, a cross-attention layer, and a feed-forward network (FFN). Specfically, the cross-attention layer makes $\mathbf{Q}$ interact with the patch tokens $\mathbf{Z}$, so as to aggregate features for locating and identifying the objects. Afterwards, the FFN transforms $\mathbf{Q}$ into the output embeddings, which make final prediction through a classification and regression head.

**DAC-DETR** follows the standard DETR pipeline for the backbone and transformer encoder and is featured for the "Divide-and-Conquer" decoders. As illustrated in Fig. 2, DAC-DETR has two decoders, *i.e.*, 1) an O-Decoder which is the same as the baseline DETR decoder, and 2) a C-Decoder that divides the cross-attention layers from the "gather $\leftrightarrow$ disperse" for better learning. C-Decoder removes all the self-attention layers and shares all the other decoder parameters (cross-attention, FFN), object queries and the prediction head with the O-Decoder. Section 3.2 elaborates on these two decoders regarding their architecture, training and inference. Section 3.3 looks into the mechanism,

*i.e.*, how DAC-DETR impacts the learning of the DETR decoder. It shows that DAC-DETR improves both the quantity (number) and the quality (larger IoU) of queries that attend to each object.

## 3.2 Divide-and-Conquer Decoders

DAC-DETR constructs two parallel decoders, *i.e.*, a canonical DETR decoder (O-Decoder) and a decoder without self-attention layers (C-Decoder), as shown in Fig. 2. The initial object queries $\mathbf{Q}$ are fed into the first block of O-Decoder and C-Decoder in parallel. Since the object queries undergo different transformations in these two decoders, they are denoted as $\mathbf{Q}_{\text{O}}^l$ (queries in O-Decoder) and $\mathbf{Q}_{\text{C}}^l$ (queries in C-Decoder), where the superscript $l(l = 0, 1, \cdots, 5)$ indicates the index of the decoder block. We have $\mathbf{Q}_{\text{O}}^0 = \mathbf{Q}_{\text{C}}^0 = \mathbf{Q}$.

• **O-Decoder: a canonical DETR-decoder revisit.** O-Decoder is a canonical DETR decoder with each block consisting of a self-attention layer, a cross-attention layer and a FFN. Specifically, the $l$-th decoder block first updates all the queries through self-attention, which is formulated as:

$$\widetilde{\mathbf{Q}}_{\text{O}}^l = \mathbf{Q}_{\text{O}}^l + \texttt{Att}_{\texttt{self}}(\texttt{Que}(\mathbf{Q}_{\text{O}}^l), \texttt{Key}(\mathbf{Q}_{\text{O}}^l), \texttt{Val}(\mathbf{Q}_{\text{O}}^l)), \tag{1}$$

where $\texttt{Que}, \texttt{Key}, \texttt{Val}$ are the projections to derive the $\texttt{query}$, $\texttt{key}$ and $\texttt{value}$ embedding from tokens $\mathbf{Q}_{\text{O}}^l$, respectively. We omit the description of "Layer norm" and "Dropout" in all equations for better clarity.

Given the queries $\mathbf{Q}_{\text{O}}^l$ output from the preceding self-attention layer, the cross-attention layer further updates them through interaction with the feature embedding $\mathbf{Z}$, which is formulated as:

$$\mathbf{Q}_{\text{O}}^{l+1} = \texttt{FFN}(\widetilde{\mathbf{Q}}_{\text{O}}^l + \texttt{Att}_{\texttt{cross}}(\texttt{Que}(\widetilde{\mathbf{Q}}_{\text{O}}^l), \texttt{Key}(\mathbf{Z}), \texttt{Val}(\mathbf{Z}))), \tag{2}$$

where $\texttt{Que}, \texttt{Key}, \texttt{Val}$ are the projections and are parameterized differently from those in the self-attention layer (Eqn. 1), though sharing the same symbols.

Finally, the output embeddings $\mathbf{Q}_{\text{O}}^{l+1}$ are fed into a predictor head to make label and position predictions. The predictions are supervised with the one-to-one matching and $\mathcal{L}_{\texttt{O-decoder}}$, which is the standard Hungarian loss [3] as in the original DETR.

• **C-decoder** removes all the self-attention layer in each decoder block and shares all the other parameters with O-decoder. Therefore, in C-Decoder, the queries undergo only cross-attention with the feature embeddings $\mathbf{Z}$ and a following FFN transformation, which is formulated as:

$$\mathbf{Q}_{\text{C}}^{l+1} = \texttt{FFN}(\mathbf{Q}_{\text{C}}^l + \texttt{Att}_{\texttt{cross}}(\texttt{Que}(\mathbf{Q}_{\text{C}}^l), \texttt{Key}(\mathbf{Z}), \texttt{Val}(\mathbf{Z}))), \tag{3}$$

where all the model parameters are shared with the O-Decoder (Eqn. 2).

In parallel to the prediction from O-Decoder, the output embeddings from C-Decoder $\mathbf{Q}_{\text{C}}^{l+1}$ are fed into the shared predictor head. The corresponding label and position prediction from a query $q \in \mathbf{Q}$ are $[p_{(q)}, b_{(q)}]$.

Different from the bipartite matching strategy in O-Decoder, C-Decoder employs one-to-many label assignment for supervision [26, 44], *i.e.*, assigning each ground-truth object with multiple queries. Specifically, given an object with ground-truth annotation $y = [\hat{c}, \hat{b}]$ ($\hat{c}$ and $\hat{b}$ are the class and bounding box, respectively), we measure its matching scores with the prediction from all the queries by:

$$\mathbf{M} = \{p_{(q)}(\hat{c}) + \texttt{IoU} < b_{(q)}, \hat{b} >\} \qquad \forall q \in \mathbf{Q}, \tag{4}$$

where $p_{(q)}(\hat{c})$ denotes the predicted label score on class $\hat{c}$, $<, >$ denotes the IoU operation between predicted box and ground truth $\hat{b}$.

Given the matching scores $\mathbf{M}$ for the ground-truth object, we select multiple positive predictions for it by two criteria: 1) the matching score should be larger than a threshold $t$ and 2) the matching score should be among the top-k scores in $\mathbf{M}$. There two criteria are required simultaneously and the second one is to suppress the label imbalance regarding different objects. All the other predictions are assigned with negative labels for this object. With the assigned labels, we supervise the predictions from C-Decoder with $\mathcal{L}_{\texttt{C-decoder}}$, which is a standard detection loss function consisting of classification [18] and regression loss [30].

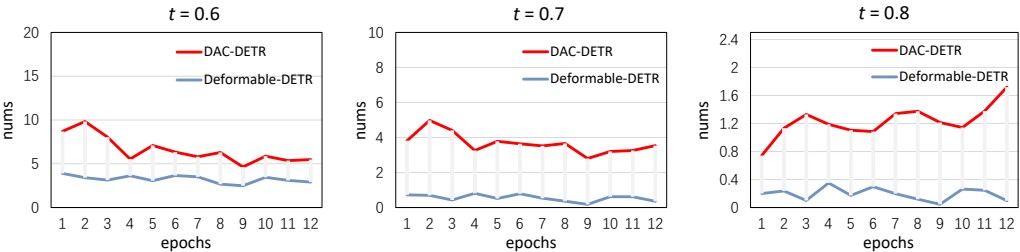

Figure 3: The averaged number of queries that each ground-truth object gathers on the validation set of MS-COCO. Compared with the Deforamble-DETR baseline, DAC-DETR 1) has more queries for each object, and 2) improves the quality of the closest queries. Results for more layers are provided in the supplementary. y-axis denotes "avg number of queries / object".

During training, $\mathcal{L}_{\text{O-decoder}}$ and $\mathcal{L}_{\text{C-decoder}}$ are summed up. During inference, DAC-DETR discards C-Decoder and maintains the same pipeline and efficiency as the baseline.

**Discussion.** Using the one-to-many label assignment for supervising C-Decoder is a natural choice. It is because C-Decoder has no self-attention layers (which are critical for removing duplicate detections) and is thus prone to multiple detections for each single object. If C-Decoder employs the one-to-one label assignment, DAC-DETR will bring no more improvement but actually compromises the baseline accuracy (Section 4.3).

### 3.3 Mechanism Analysis

We investigate how the proposed divide-and-conquer method benefits DETR with focus on the gathering effects of the cross-attention layers. Specifically, we are interested in: *how many queries can each ground-truth object gather in DAC-DETR and in the baseline?* To this end, we count the averaged number of queries that have large affinity (according to the matching score in Eqn. 4) with each object. The queries with high matching scores are important: they aggregate abundant information from a corresponding object and thus serve as the proposals which are potential to predict the object. Based on these highly-matched queries (proposals), the self-attention layer removes redundant queries and gives the final non-duplicate prediction.

The comparison between DAC-DETR and the Deformable-DETR baseline is in Fig. 3. We list the results for the last decoder layer and leave the results for more layers to the supplementary. From Fig. 3, we draw two important observations as below:

• *Remark-1: DAC-DETR increases the number of queries gathered for each object.* Comparing DAC-DETR and its baseline under various matching score threshold (*i.e.*, $t = 0.6, 0.7, 0.8$), we observe that DAC-DETR consistently gathers more queries for each single object. Please note that the total number of queries is the same for DAC-DETR and the baseline. We thus infer that DAC-DETR makes higher utilization of the object queries.

• *Remark-2: DAC-DETR improves the quality of the best queries for each object.* Focusing on the comparison under large threshold (*e.g.*, $t = 0.8$), we find the baseline has rare high-quality queries, while DAC-DETR has about 1 high-quality query for each object. We recall that the matching score is composed of IoU and classification score. This suggests that some queries in DAC-DETR either has good IoU or high confidence to the ground-truth object.

Combining the above two remarks, we conclude that DAC-DETR improves both the quantity and quality of the queries for each object. This improvement is the direct reason how DAC-DETR increases the detection accuracy (Section 4).

## 4 Experiments

### 4.1 Setup

**Dataset and backbone.** We evaluate the proposed DAC-DETR on COCO 2017 [17] detection dataset. Following the common practices, we evaluate the performance on validation dataset(5k images)

| Method | Backbone | epochs | AP | AP$_{50}$ | AP$_{75}$ | AP$_S$ | AP$_M$ | AP$_L$ |
|---|---|---|---|---|---|---|---|---|
| Basel#1 (Deformable) [45] | R50 | 12 | 43.7 | 62.0 | 47.3 | 26.2 | 46.8 | 58.1 |
| Basel#1 (Deformable) [45] | R50 | 36 | 46.8 | 66.0 | 50.6 | 29.4 | 50.1 | 61.4 |
| H-DETR [12] | R50 | 12 | 45.9 | - | - | - | - | - |
| DN-Deformable-DETR [15] | R50 | 12 | 46.0 | 63.8 | 49.9 | 27.7 | 49.1 | 62.3 |
| DAC-DETR (ours) | R50 | 12 | 47.1 (+ 3.4) | 64.8 | 51.1 | 29.2 | 50.6 | 62.4 |
| DAC-DETR (ours) | R50 | 24 | 48.5 (+ 1.7)* | 66.5 | 52.6 | 31.0 | 51.8 | 63.1 |
| Basel#2 (Deformable++) [12] | R50 | 12 | 47.6 | 65.9 | 52.3 | 30.2 | 51.1 | 62.6 |
| Basel#2 (Deformable++) [12] | R50 | 36 | 49.3 | 67.5 | 53.6 | 32.1 | 52.5 | 64.4 |
| H-DETR [12] | R50 | 12 | 48.7 | 66.4 | 52.9 | 31.2 | 51.5 | 63.5 |
| H-DETR [12] | R50 | 36 | 50.0 | 68.3 | 54.4 | 32.9 | 52.7 | 65.3 |
| DAC-DETR (ours) | R50 | 12 | 49.3 (+1.7) | 66.5 | 53.8 | 31.4 | 52.4 | 64.1 |
| DAC-DETR (ours) | R50 | 24 | 50.5 (+1.2)* | 67.9 | 55.2 | 33.2 | 53.5 | 64.8 |
| Basel#3 (DINO) [42] | R50 | 12 | 49.0 | 66.6 | 53.5 | 32.0 | 52.3 | 63.0 |
| Basel#3 (DINO) [42] | R50 | 24 | 50.4 | 68.3 | 54.8 | 33.3 | 53.7 | 64.8 |
| Group-DETR [5] | R50 | 12 | 49.8 | - | - | 32.4 | 53.0 | 64.2 |
| Align-DETR [1] | R50 | 12 | 50.2 | 67.8 | 54.4 | 32.9 | 53.3 | 65.0 |
| Align-DETR [1] | R50 | 24 | 51.3 | 68.2 | 56.1 | **35.5** | 55.1 | 65.6 |
| Stable-DINO-4scale [20] | R50 | 12 | 50.4 | 67.4 | 55.0 | 32.9 | 54.0 | 65.5 |
| Stable-DINO-4scale [20] | R50 | 24 | 51.5 | 68.5 | 56.3 | 35.2 | 54.7 | 66.5 |
| DAC-DETR (ours) | R50 | 12 | 50.0 (+1.0) | 67.6 | 54.7 | 32.9 | 53.1 | 64.2 |
| DAC-DETR (ours) | R50 | 24 | 51.2 (+0.8) | 68.9 | 56.0 | 34.0 | 54.6 | 65.4 |
| DAC-DETR + Align (ours) | R50 | 12 | 50.9 (+1.9) | 68.3 | 55.3 | 33.8 | 54.1 | 65.7 |
| DAC-DETR + Align (ours) | R50 | 24 | **52.1** (+1.7) | **69.7** | **56.5** | 34.8 | **55.2** | **67.3** |

Table 1: Evaluation on COCO val2017 with ResNet-50 backbone . For fair comparison, the methods are divided into three groups according to their baseline , *i.e.*, Basel#1: Deformable-DETR [45], Basel#2: Deformable-DETR++ [12], Basel#3: DINO [42]. *: We compare performance between 24-epoch DAC-DETR and 36-epoch baseline.

| Method | Backbone | epochs | AP | AP$_{50}$ | AP$_{75}$ | AP$_S$ | AP$_M$ | AP$_L$ |
|---|---|---|---|---|---|---|---|---|
| H-DETR [12] | Swin-L | 12 | 56.1 | 75.2 | 61.3 | 39.3 | 60.4 | 72.4 |
| H-DETR [12] | Swin-L | 36 | 57.6 | 76.5 | 63.2 | 41.4 | 61.7 | 73.9 |
| Basel#3 (DINO) [42] | Swin-L | 12 | 56.8 | 75.6 | 62.0 | 40.0 | 60.5 | 73.2 |
| Basel#3 (DINO) [42] | Swin-L | 36 | 58.0 | 77.1 | **66.3** | 41.3 | 62.1 | 73.6 |
| Group-DETR [5] | Swin-L | 36 | 58.4 | - | - | 41.0 | 62.5 | 73.9 |
| Stable-DINO-4scale [20] | Swin-L | 12 | 57.7 | 75.7 | 63.4 | 39.8 | 62.0 | 74.7 |
| Stable-DINO-4scale [20] | Swin-L | 24 | 58.6 | 76.7 | 64.1 | 41.8 | 63.0 | 74.7 |
| DAC-DETR (ours) | Swin-L | 12 | 57.3 (+0.5) | 75.7 | 62.7 | 40.1 | 61.5 | 74.4 |
| DAC-DETR + Align (ours) | Swin-L | 12 | 58.1 (+1.3) | 76.5 | 63.3 | 40.9 | 62.4 | 75.0 |
| DAC-DETR + Align (ours) | Swin-L | 24 | **59.2** (+1.2)* | **77.1** | 64.5 | **43.1** | **63.4** | **76.0** |

Table 2: Evaluation on COCO val2017 with Swin-Transformer Large backbone. Most competing methods use the same baseline, *i.e.*, DINO [42]. *: We compare performance between 24-epoch DAC-DETR and 36-epoch baseline.

by using standard average precision (AP) result under different IoU thresholds. We implement DAC-DETR with two popular backbones, *i.e.*, ResNet50 [10] (pretrained on ImageNet-1k [7]) and Swin-Large [21] (pretrained on ImageNet-22k [7]).

**Baseline and implementation details.** The baselines for recent state-of-the-art methods vary a lot. For fair comparison with these methods, we adopt three recent popular baselines, *e.g.*, the original Deformable-DETR [45], an improved Deformable DETR (Deformable++) [12] and the DINO [42] baseline. As for training, we use AdamW [22, 14] optimizer with weight decay of $1 \times 10^{-4}$. We report experimental results under the $1\times$ (12 epochs) and $2\times$ (24 epochs) scheme. We find the $2\times$ learning scheme already allows DAC-DETR to achieve superior accuracy, compared with most competing methods (some of which uses longer training epochs).

## 4.2 Main Results

We evaluate DAC-DETR on COCO 2017 detection validation dataset. The results on ResNet50 [10] and Swin-Transformer Large backbones [21] are summarized in Table 1 and Table 2, respectively.

| Method | Backbone | epochs | AP | AP$_{50}$ |
|---|---|---|---|---|
| Baseline (Deformable-DETR [45]) | R50 | 12 | 43.7 | 63.0 |
| Variant1 (freeze cross-atten layers in O-Decoder) | R50 | 12 | 45.9 | 63.3 |
| Variant2 (add self-atten layers in C-Decoder) | R50 | 12 | 43.1 | 58.9 |
| Variant3 (one-to-one matching in C-Decoder) | R50 | 12 | 43.5 | 62.1 |
| DAC-DETR (ours) | R50 | 12 | 47.1 | 64.8 |

Table 3: Comparison of four variants for DAC-DETR.

| Method | Backbone | epochs | AP | AP$_{50}$ | AP$_{75}$ |
|---|---|---|---|---|---|
| Basel#1 (Deformable) [45] | R50 | 12 | 43.7 | 63.0 | 47.6 |
| DAC-DETR w/ one-to-many Hungarian | R50 | 12 | 45.5 | 64.0 | 49.3 |
| DAC-DETR (ours) | R50 | 12 | 47.1 | 64.8 | 51.1 |
| Basel#2 (Deformable++) [12] | R50 | 12 | 47.0 | 65.2 | 51.5 |
| DAC-DETR w/ one-to-many Hungarian | R50 | 12 | 48.4 | 65.6 | 52.8 |
| DAC-DETR (ours) | R50 | 12 | 49.3 | 66.5 | 53.8 |
| Basel#3 (DINO) [42] | R50 | 12 | 49.0 | 66.6 | 53.5 |
| DAC-DETR w/ one-to-many Hungarian | R50 | 12 | 49.3 | 66.7 | 53.8 |
| DAC-DETR (ours) | R50 | 12 | 50.0 | 67.6 | 54.7 |

Table 4: Comparison of different one-to-many matching strategies for DAC-DETR. Hungarian-based one-to-many matching is useful but is inferior to our threshold-based strategy.

Based on the ResNet50 backbone, we employ three different baselines in Table 1 ( with ResNet50 backbones ), from which we draw two observations: **1)** DAC-DETR achieves consistent and remarkable improvements over all three baselines. For example, under $1\times$ learning scheme, DAC-DETR improves Deformable-DETR [45], Deformable-DETR++ [12] and DINO [42] by +3.4, +2.3 and +1.0 AP, respectively. **2)**, DAC-DETR achieves competitive detection accuracy. We note that recent state-of-the-art methods usually adopt a strong baseline (*e.g.*, DINO) and further integrate a battery of good practices. Among these methods, Stable-DINO is the strongest and combines an IoU-related loss, an improved matching strategy and a novel feature fusion. On the same baseline (DINO), our method (DAC-DETR + Align) , while employing fewer tricks (only Align Loss), surpasses Stable-DINO by + 0.5 AP (12 epochs) and +0.6 AP (24 epochs).

On the Swin-L backbone (where relatively fewer methods have reported results), we only adopt the strongest baseline (DINO) and compare DAC-DETR against most recent state-of-the-art methods. The results in Table 2 further confirms the effectiveness of DAC-DETR: it improves DINO baseline by +0.5 AP under 12 epochs. After combining an IoU-related loss (Align [1] loss), DAC-DETR surpasses the strongest competitor Stable-DINO by + 0.4 AP (12 epochs) and +0.6 AP (24 epochs).

### 4.3 Ablation Studies

**Three variants of DAC-DETR**. In Table 3, we are interested in two questions: 1) Since the cross-attention layers already undergo specific training in C-Decoder, can they be frozen in O-Decoder (Variant-1)? 2) Are removing self-attention layers and using one-to-many matching important for C-Decoder? In response, we implement Variant-2&3, which adds the self-attention layers and performs one-to-one matching for C-Decoder, respectively. From Table 3, we draw the two following observations:

First, freezing the cross-attention layers in O-Decoder considerably compromises DAC-DETR (-1.2 AP), but still improves the baseline (+2.2). It shows that training the cross-attention layers only in C-Decoder already suffices a competitive DETR detector and brings major improvement. We thus infer that C-Decoder makes the major contribution to learning the cross-attention layers. On the O-Decoder side, it pays relatively fewer efforts onto cross-attention layers and channels its efforts onto learning the self-attention layers. Second, adding self-attention and using one-to-one matching for C-Decoder both deteriorate DAC-DETR. These two variants are even inferior to the baseline by -0.6 and -0.2 AP respectively. We thus infer that removing the self-attention layers and using

| Method | Backbone | epochs | AP | $AP_{50}$ |
|---|---|---|---|---|
| Deformable-DETR [45] | R50 | 12 | 43.7 | 63.0 |
| DAC-DETR w/o regression loss in C-decoder | R50 | 12 | 45.7 | 64.0 |
| DAC-DETR w/o sync position | R50 | 12 | 46.8 | 64.3 |
| DAC-DETR (full) | R50 | 12 | 47.1 | 64.8 |

Table 5: Investigation on some details for learning the query position.

| Method | Backbone | queries | AP |
|---|---|---|---|
| Deformable-DETR [45] | R50 | 300 | 43.7 |
| H-DETR (our implementation) | R50 | 300 + 1500 | 46.4 |
| H-DETR w/o self-atten in one-to-many decoder | R50 | 300 + 1500 | 47.0 |
| DAC-DETR | R50 | 300 | 47.1 |

Table 6: Removing the self-attention in the one-to-many decoder improves H-DETR, as well.

one-to-many matching are critical. Being featured for no self-attention layers, C-Decoder has no capability to remove duplicate predictions and thus favors one-to-many matching.

**Comparison of different one-to-many matching strategies**. In DAC-DETR, C-Decoder adopts a threshold-based one-to-many matching strategy. We investigate another alternative, *i.e.*, Hungarian matching with repeated ground-truth in H-DETR [12] in Table 4 and draw two observations:

First, comparing "DAC-DETR w/ one-to-many Hungarian" against the baseline, we observe consistent improvement. It shows that C-Decoder is compatible to Hungarian-based one-to-many matching, as well. Second, comparing two one-to-many strategies against each other, we observe our threshold-based strategy is better. This is reasonable because Hungarian-based one-to-many matching is based on set-to-set comparison and requires global comparison among all the samples. C-Decoder has no self-attention layers for global comparison and thus favors the threshold-based strategy.

**Some details for learning the query position**. Table 5 investigates some training details *w.r.t.* the query position. It is observed that removing the regression loss for C-Decoder achieves relatively small improvement. It is consistent with our motivation in Fig. 1, *i.e.*, the gathering effect concerns the position as well. Moreover, there is a training trick, *i.e.*, given a same query input, we synchronize its position in O-Decoder and C-Decoder through average. This operation brings +0.3 AP improvement on DAC-DETR.

**Can removing self-attention benefit other one-to-many decoders?** Table 3 (Variant2) shows removing self-attention is critical for C-Decoder. Table 6 further investigates this design on another popular one-to-many decoder, H-DETR [12] and shows non-trivial improvement (+0.6AP) in 12 training epochs, as well. It further confirms that removing the self-attention enhances the effect of gathering multiple queries towards each object on different one-to-many decoders.

## 5 Conclusion

This paper reveals in the DETR decoder, the cross-attention and self-attention layers have opposing impacts (*i.e.*, gathering and dispersing) on the object queries. This insight motivates us to divide the cross-attention layers out from this opposition for better conquering. The corresponding method, DAC-DETR, improves both the quantity and quality of the queries that are gathered by each object. Experimental results show that DAC-DETR improves multiple popular DETR baselines (*e.g.*, Deformable-DETR, DINO) and performs favorably against recent state-of-the-art methods.

**Acknowledgements**. This work is supported by the Fundamental Research Funds for the Central Universities (No. 226-2023-00048).

**Broader Impacts**. This paper utilizes the open dataset to evaluate the performance. Our method can improve the object detection, which could be applied to automatic driving systems. We will enhance the generalization of method.

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
