# DAC-DETR: Divide the Attention Layers and Conquer

## A Supplementary Material

### A.1 Gathering effect on more decoder layers

Section 3.3 (Mechanism Analysis) in the main text shows that DAC-DETR improves the gathering effect of the cross-attention layer, *i.e.* more and better queries (Fig. 3 in the main text). We supplement results on more decoder layers (layer-2, layer-3 and layer-4) in Fig. A1.

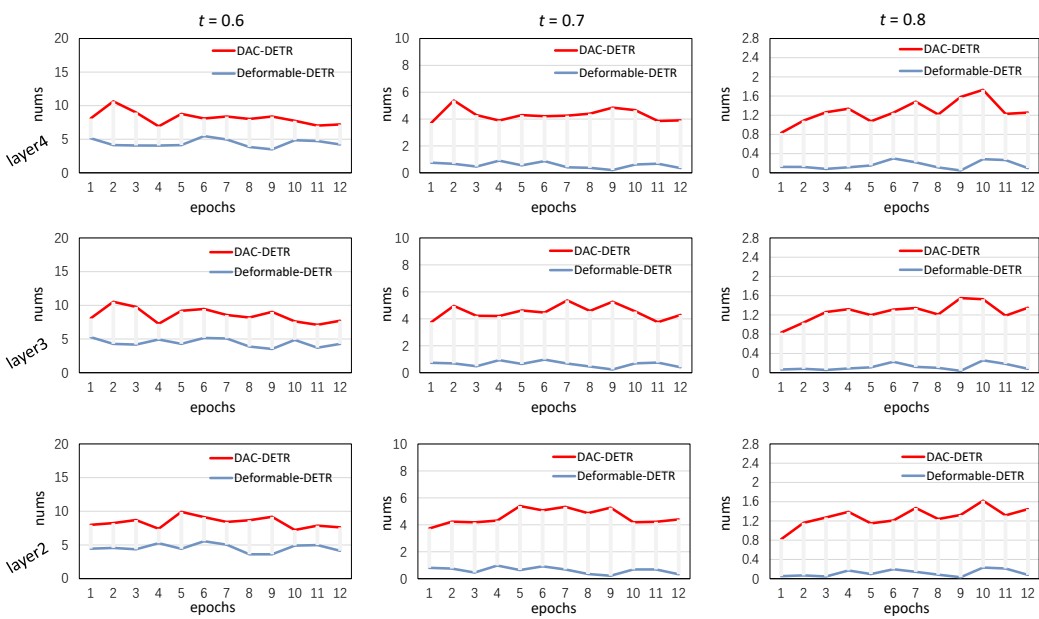

Figure A1: The averaged number of queries that each ground-truth object gathers on the validation set of MS-COCO.

The observation in Fig. A1 is consistent with Fig. 3 in the main text: DAC-DETR gathers more queries for each single object and improves the quality of the best queries. The two corresponding remarks, *i.e.*, DAC-DETR improves the quantity and quality of the gathered queries, hold across multiple decoder layers.

### A.2 Comparison on Convergence Speed

We investigate the convergence speed of DAC-DETR on three baselines (Deformable-DETR [8] , Deformable-DETR++ [3], and DINO [7] ) in Fig. A2. The experiments are conducted on the COCO 2017 [4] detection validation dataset. We adopt ResNet50 [2] backbone and run 12 epochs. It is observed that DAC-DETR consistently improves the convergence speed over all three baselines. For example, DAC-DETR outperforms the Deformable-DETR baseline by +8.3 AP and +3.4 AP at epoch-1 and epoch-12, respectively.

Submitted to 37th Conference on Neural Information Processing Systems (NeurIPS 2023). Do not distribute.

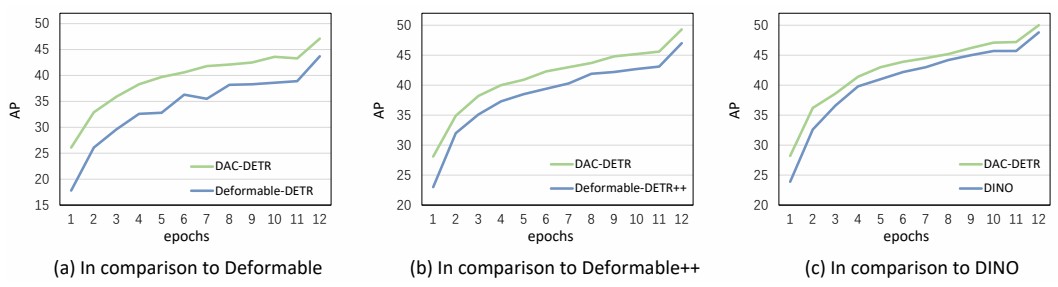


(a) In comparison to Deformable     (b) In comparison to Deformable++     (c) In comparison to DINO


Figure A2: Comparison of convergence speed between DAC-DETR and three baselines.

## A.3 Comparison of training time and inference FPS

We compare the average training time per epoch and the inference FPS between DAC-DETR, H-DETR [3], and baseline method (Deformable-DETR [8] ). For a fair comparison, all the methods utilize 8 A100 GPUS for training and a single A100 GPU for inference.

| Method | Backbone | Training time (average) | Inference FPS | AP |
|---|---|---|---|---|
| Basel (Deformable) [8] | R50 | 58 min | 17.8 | 43.7 |
| H-DETR [3] | R50 | 70 min | 17.8 | 45.9 |
| DAC-DETR (ours) | R50 | 64 min | 17.8 | 47.1 |

Table A1: Comparison of average training time on each epoch and inference FPS.

From Table A1, we draw two observations: 1) Compared to the baseline, DAC-DETR increases the training time per epoch by a small margin (*i.e.*, +6 minutes) while maintaining the same inference efficiency. The small increase on training time is because DAC-DETR additionally introduces an auxiliary decoder (*i.e.*, C-Decoder) that processes all the queries in parallel. 2) Compared with H-DETR (a recent method that employs auxiliary decoder branch), our DAC-DETR is faster to train (-6 minutes per epoch). There are two reasons: first, DAC-DETR uses fewer queries than H-DETR. Second, the auxiliary C-Decoder in DAC-DETR has fewer attention layers (*i.e.*, no self-attention layers).

## A.4 More hyper-parameter analysis

In our one-to-many label assignment (Eqn.4 in the main text), we compute the matching score $m$ between each query and the object by adding their IoU score and the predicted label score on the ground-truth class. To introduce more flexibility, we combine these two scores through a weighted sum, which is formulated as:

$$m = (1 - \lambda) \cdot p_{(q)}(\hat{c}) + \lambda \cdot \texttt{IoU} < b_{(q)}, \hat{b} >, \tag{1}$$

where $\lambda$ is a newly-added hyper-parameter for weighting, and all the other variables are the same as in the main text (*i.e.*, $\hat{c}$ and $\hat{b}$ are the class and bounding box of query $q$, $p_{(q)}(\hat{c})$ denotes the predicted label score on class $\hat{c}$. $b_{(q)}$ denotes the predicted box, $<,>$ denotes the IoU operation between predicted box and ground truth $\hat{b}$). We investigate the influence of this hyper-parameter $\lambda$ in Table A2.

| $\lambda$ | 0.5 | 0.6 | 0.7 | 0.8 | 0.9 | 1.0 |
|---|---|---|---|---|---|---|
| AP | 46.8 | 47.0 | 47.1 | 47.0 | 46.8 | 46.6 |

Table A2: Analysis on the weight $\lambda$ in the one-to-many label assignment. We adopt Deformable-DETR as the baseline.

We observe that DAC-DETR is robust to this hyper-parameter within a large range, and in practice use $\lambda = 0.7$ for all the experiments.

## A.5    More Experiments

We evaluate the performance of Align-DETR [1] with the Swin-L [6] backbone on COCO 2017 detection validation dataset, using the official publicly codes. ( Align-DETR does not report the results with Swin-L backbone). The results in Table A3 further confirms the superiority of DAC-DETR. After combining an IoU-related loss (Align loss), DAC-DETR surpasses Align-DETR by +0.7 AP (12 epochs).

| Method | Backbone | epochs | AP | $AP_{50}$ | $AP_{75}$ | $AP_S$ | $AP_M$ | $AP_L$ |
|---|---|---|---|---|---|---|---|---|
| Basel (DINO) [7] | Swin-L | 12 | 56.8 | 75.6 | 62.0 | 40.0 | 60.5 | 73.2 |
| Align-DETR † [1] | Swin-L | 12 | 57.4 | 75.9 | 62.2 | 40.6 | 61.6 | 73.7 |
| Stable-DINO-4scale [5] | Swin-L | 12 | 57.7 | 75.7 | 63.4 | 39.8 | 62.0 | 74.7 |
| DAC-DETR + Align (ours) | Swin-L | 12 | 58.1 | 76.5 | 63.3 | 40.9 | 62.4 | 75.0 |

Table A3: Evaluation on COCO val2017 with Swin-Transformer Large backbone. †: We evaluate Align-DETR using the official publicly codes.

## A.6    Visualization of Object Detection

We visualize some detection results with predicted bounding boxes and label scores in Fig. A3 and Fig. A4. As shown in Fig. A3, DAC-DETR detects the object "zebra" with limited semantic information, whereas Deformable-DETR fails to do so. Compared to Deformable-DETR++, DAC-DETR provides more accurate label and box predictions for the object "cat", as shown in Fig. A4.

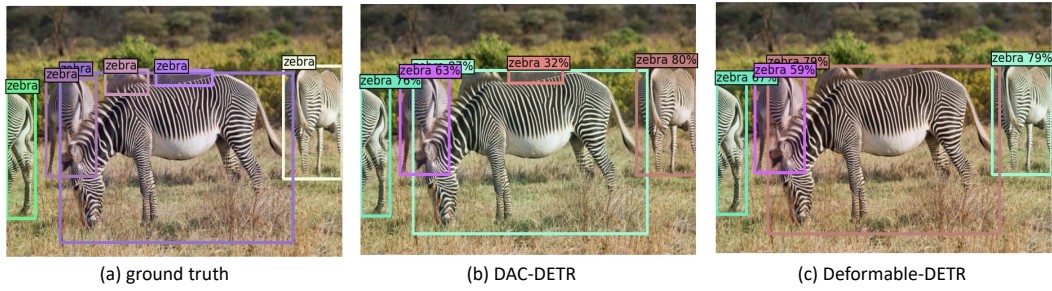

(a) ground truth            (b) DAC-DETR            (c) Deformable-DETR

Figure A3: Visualization of the detection results of DAC-DETR and Deformable-DETR.

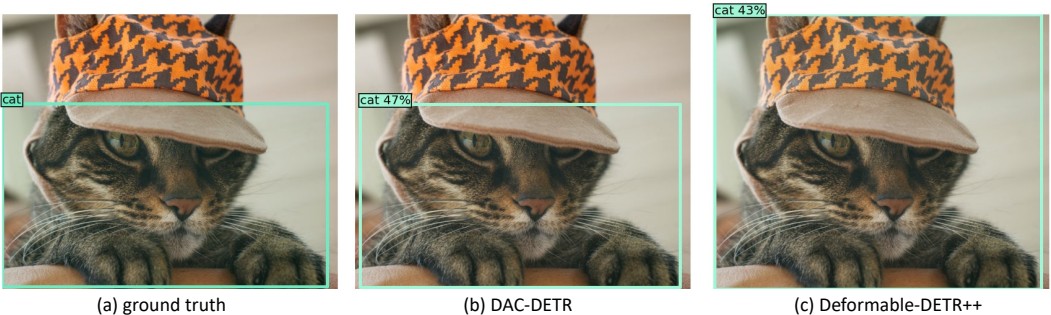

(a) ground truth            (b) DAC-DETR            (c) Deformable-DETR++

Figure A4: Visualization of the detection results of DAC-DETR and Deformable-DETR++.