# OpenReview forum: "DAC-DETR: Divide the Attention Layers and Conquer"
_NeurIPS.cc/2023/Conference — NeurIPS 2023 poster_

### Official Review · Reviewer_VXuS · 2023-07-02

**Soundness:** 4 excellent
**Presentation:** 4 excellent
**Contribution:** 3 good
**Rating:** 7
**Confidence:** 5

**Summary:**

This paper introduces DAC-DETR, a method to improve the training efficacy of DEtection Transformers (DETR) by addressing the contrary impacts of cross-attention and self-attention layers in the DETR decoder. DAC-DETR divides the cross-attention layers into an auxiliary decoder, which focuses on learning the cross-attention, while the original decoder handles non-duplicate detection. By employing one-to-many label assignment in the auxiliary decoder, DAC-DETR effectively improves the gathering effect of queries, leading to improved detection accuracy compared to popular DETR baselines.

**Strengths:**

- The paper is well-organized, presenting its ideas and findings in a clear and coherent manner.
- The motivation is clear. The proposed methods are straightforward and logical, making them easy to understand and implement.
- The effectiveness of the proposed methods is well-supported by a comprehensive set of experiments. The experimental results demonstrate significant improvements over existing DETR models.

**Weaknesses:**

I have no severe concerns about the paper.

**Questions:**

The reviewer is curious about the performance of the one-to-many branch (C-decoder).

---

> ### Author Rebuttal · Authors · 2023-08-10
>
> **VXuS-Q1: The reviewer is curious about the performance of the one-to-many branch (C-decoder).**
>
> **[Ans]:** Thanks for this question. C-Decoder (with NMS) achieves slightly lower accuracy than the O-Decoder. Since C-Decoder has no self-attention layers and makes duplicate detection, NMS is prerequisite for filtering out the duplicate predictions. Based on the Deformable-DETR baseline (43.7 AP), using C-Decoder (+NMS) achieves 46.9 AP, which is -0.2 lower than using O-Decoder (47.1 AP).
>
> Moreover, we notice a related question (raised by review **LpXx-Q3**) that might draw your interest as well: combining the predictions from C-Decoder and O-Decoder may further bring incremental improvement. For example, on the above setting, combining the C-Decoder (46.9 AP) and O-Decoder (47.1 AP) achieves 47.4 AP. Please kindly refer to **LpXx-Q3** for more details.

---

### Official Review · Reviewer_QUSJ · 2023-07-04

**Soundness:** 3 good
**Presentation:** 3 good
**Contribution:** 3 good
**Rating:** 7
**Confidence:** 4

**Summary:**

This paper observes the problems in cross-attention and self-attention that impacts the queries, and proposes to use divide-and-conquer to improve the training accuracy.

**Strengths:**

1. The paper is considered novel to me. The insights in analyzing the cross-attention and self-attention and the proposed design are interesting.
2. Strong performance. The final performance is improved compared with SOTA detectors.

**Weaknesses:**

1. The design seems a little complex. Not sure if this design can be plugged into other DETR-like model easily.

**Questions:**

No

---

> ### Author Rebuttal · Authors · 2023-08-10
>
> **QUSJ-Q1: The design seems a little complex. Not sure if this design can be plugged into other DETR-like model easily.**
>
> **[Ans]:** Thanks for this question. Plugging our method into the DETR-like models is easy. Given a DETR-like model, we only need to append a C-Decoder with two steps: **1)** replicating its original decoder and **2)** removing the self-attention layers. The C-Decoder is supervised with the one-to-many matching, which may seem a bit complex but is reliable (DAC-DETR has considerable robustness on the one-to-many hyper-parameters, as shown in Fig. 4 in the manuscript). In our experiments, we already plug our DAC-DETR into multiple popular baselines, *i.e.*, Deformable-DETR, Deformable-DETR++, DINO, and achieve consistent improvement. We show our method is compatible to several most recent good practices, *e.g.*, Aligned loss and stable matching (during rebuttal), as well. In all these experiments, we use the same hyper-parameters for DAC-DETR.

---

> > ### Comment · Reviewer_QUSJ · 2023-08-15
> > **Thanks for the rebuttal.**
> >
> > Thanks for the rebuttal. I also notice that the final best performance is achieved by '+align', which is adopted from another paper Align DETR. Could the proposed method improve the final performance or just accelerate convergence? From the table, it is still unclear.

---

> > > ### Author Response · Authors · 2023-08-16
> > > **Response to Reviewer QUSJ**
> > >
> > > Thanks. After your kind reminder, we find that our comparison indeed needs some re-organization and highlight to be more clear. DAC-DETR's benefits include both faster convergence and higher final accuracy.
> > >
> > > An supporting observation is in Table 1 in the manuscript. Specifically, based the DINO (ResNet-50) baseline, DAC-DETR (**NO** align loss) brings +1.0 (49.0 $\rightarrow$ 50.0), +0.8 (50.4 $\rightarrow$ 51.2) and +0.6 (50.9 $\rightarrow$ 51.5) under 12epochs, 24 epochs, and 36 epochs, respectively. The results under 12 and 24 epochs are already provided in Table 1 in the manuscript, and we will add the 36-epoch results into Table 1.
> > > It shows that:
> > >
> > > * The improvement under short training schedule is larger (+1.0 AP), indicating faster convergence.
> > > * The non-trivial improvement (+0.6AP) still holds when the training schedule is long (*i.e.*,36 epochs).
> > >
> > >
> > >
> > > Moreover, if we consider Align DETR as another baseline, the non-trivial improvements still hold. For example, using the Swin-L backbone, DAC-DETR improves Align DETR by +0.7 (57.4 $\rightarrow$ 58.1) and +1.0 (58.2 $\rightarrow$ 59.2) under 12 and 24 epochs, respectively. The baseline results of Align DETR are not listed in the main text and are provided in the Appendix (Table A3). We will add the comparison to Table 2 in the main text.

---

### Official Review · Reviewer_CkgD · 2023-07-06

**Soundness:** 3 good
**Presentation:** 2 fair
**Contribution:** 2 fair
**Rating:** 4
**Confidence:** 4

**Summary:**

The authors find that the cross-attention and self-attention in the DETR decoder have opposite effects on object queries. This phenomenon reduces the training efficiency of DETR model. To resolve the contradiction, this paper proposes a Divide-And-Conquer DETR that employs an auxiliary decoder that shares parameters with the original decoder. Experimental results show that DAC-DETR achieves significant performance over popular DETRs.

**Strengths:**

The paper adequately cites relevant work and clearly articulates the differences from existing work. The experiments show promising results. And other researchers may build a network structure based on this paper.

**Weaknesses:**

* *Originality**: This article is a combination of a series of existing methods and is not innovative enough.
* *Quality*：The motivation is straightforward, but the authors should give more analysis about contradiction. DAC-DETR brings improvement over baseline, but this does not indicate that the problem has been solved by theoretical analysis or experimental results.
* *Clarity*：This paper is not well organized and has some grammar errors.
* *Significance*：The model architecture is not adequately simple, which will affect its usability. From the experimental results, the authors proposed DAC-DETR to solve the conflict problem of cross-attention and self-attention but did not fundamentally innovate the structure to solve the problem.

**Questions:**

1.In Figure 1 (b,c), which module is used to obtain the classification score and bounding box of the object? Why "cls" of objects in Figure 1(b) is so different from "cls" scores in Figure 1(b). Does the same phenomenon occur at the last layer of the DETR decoder and occur at the DINO?
2. This paper proposes DAC-DETR to resolve the contradiction between cross-attention and self-attention. Whether this problem in Figure1 is solved by the corresponding method, the authors should give more visualization analysis like Figure1, used DAC or not.
3. In Table 3. why the AP of varaiant2 and varaiant3 is reduced?

**Limitations:**

1. The description of the one-to-many label for assignment In section 3.2 is a bit too abstract. This is a key component of the proposed method, from Table 4, and the major difference with the existing method.
2. The authors proposed DAC-DETR to solve the confliction problem of cross-attention and self-attention, but innovative structures are not proposed to fundamentally solve the problem.
3. Compared with one-to-many Hungarian Loss in Table 4, the increase of AP is due to matching scores introducing an IoU score, which is also compatible with the original DETR.

---

> ### Author Rebuttal · Authors · 2023-08-10
>
> **CkgD-Q1: This article is a combination of a series of existing methods and is not innovative enough.**
>
> **[Ans]:**  We respectfully disagree with this point. We would like to highlight our major contributions as below:
>
> 1) We reveal a characteristic of DETR, *i.e.*, the cross-attention and self-attention layers in DETR decoder have opposing effects of "gather $\leftrightarrow$ disperse". This characteristic has not been noticed before and provides a new insight for understanding the training difficulty of DETR.
>
>
> 2) Based on this insight, we propose to separate the cross-attention out from the contradiction. This objective is implemented with a simple design, *i.e.*, adding a C-Decoder that replicates the original decoder but removes all the self-attention layers.
>
> 3) We show that DAC-DETR brings consistent improvements to popular DETR-like methods. For example, under 12 epochs learning scheme, DAC-DETR improves Deformable-DETR, Deformable-DETR++ and DINO by +3.4, +2.3 and +1.0 AP, respectively.
>
> * * *
>
> **CkgD-Q2: The motivation is straightforward, but there should be analysis (in addition to the accuracy improvement) to show the problem has been solved.**
>
> **[Ans]:** Thanks. Section 3.3 (mechanism analysis) and Fig. 3 in the manuscript have already shown that DAC-DETR suppresses the contradiction: **1)** the number of queries gathered to each object is enlarged, and **2)** the best query for each object becomes even closer to the corresponding object.
>
> * * *
>
> **CkgD-Q3: The model architecture is not adequately simple, which affects its usability.**
>
> **[Ans]:** We respectfully recall that our method only adds small complexity (C-Decoder) to the baseline structure. C-Decoder is simple: it replicates the structure of the original decoder (O-Decoder) and removes the self-attention layers. Our architecture complexity is comparable (usually lower), compared with recent state-of-the-art methods, *e.g.*, Hybrid-DETR adds a parallel decoder branch that have more quires, Group-DETR replicates the original decoder for multiple times. Correspondingly, our DAC-DETR has higher training efficiency compared with recent state-of-the-art methods (Section A.3 in the appendix).
>
> * * *
>
> **CkgD-Q4: In Fig.1 (b,c), which module is used to obtain the classification score and bounding box of the object? Why "cls" of objects in Fig.1(b) is different from "cls" scores in Fig.1 (c). Does the same phenomenon occur at the last layer of the DETR decoder and occur at the DINO?**
>
> **[Ans]:** Thanks for the questions. **1)** The classification score and the bounding box prediction are obtained at the last (6-th) decoder layer. **2)** The classification score difference is because in Fig.1 (b) removes all the self-attention layers in the decoder of the already-trained model in Fig.1 (c). Correspondingly, the detector loses the capability to suppress duplicates (L36 in the manuscript) and thus increases the predicted score for multiple queries. **3)** Fig. 1 already uses the last layer. The phenomenon is similar in earlier layers and also occurs in DINO (as visualized in the supplementary PDF for rebuttal).
>
> * * *
>
> **CkgD-Q5: Analysis on whether the contradiction problem in Figure1 is solved by the corresponding method.**
>
> **[Ans]:** Thanks. Section 3.3 (mechanism analysis) and Fig. 3 in the manuscript have already shown that DAC-DETR suppresses the contradiction. The evidences are **1)** the number of queries gathered to each object is enlarged and **2)** the best query for each object becomes even closer to the corresponding object. We note that Fig. 3 is a statistic across the whole validation set and is more general than single sample visualization.
>
> * * *
>
> **CkgD-Q6: In Table 3. why the AP of variant-2\&3 is reduced?**
>
> **[Ans]:** Thanks for this question. We guess your question is why the AP of variant-2 and variant-3 is even slightly lower than the baseline. This is reasonable. **1)** Variant-2 re-adds the self-attention layers into the C-Decoder, and thus can be viewed as having two O-Decoders. Applying additional one-to-many matching for the O-Decoder makes the model prone to duplicate predictions and thus reduces the accuracy. **2)** Variant-3 uses one-to-one matching for C-Decoder, while C-Decoder has no self-attention and is naturally prone to make duplicate (one-to-many) predictions. In other words, in Variant-3, the supervision has some conflict with the characteristic of C-Decoder (L255 in the manuscript), therefore reducing the accuracy.
>
> * * *
>
> **CkgD-Q7: The description of the one-to-many matching in section 3.2 is a bit too abstract. This is a key component of the proposed method, from Table 4, and the major difference with the existing method.**
>
> **[Ans]:** We respectfully disagree. The one-to-many label assignment is only a technical detail for training our C-Decoder and is indeed simple: we calculate the matching score (Eqn. 4) and assign the positive labels to highly-scored queries. Our core contributions are the discovery of the opposing "gather $\leftrightarrow$ disperse" phenomenon and the corresponding C-Decoder solution. The C-Decoder is not restricted to any particular one-to-many method and is potential to utilize better one-to-many method for further improvement.
>
> * * *
>
> **CkgD-Q8: In Table 4, compared with one-to-many Hungarian Loss, the increase of AP is due to your matching score introducing an IoU score.**
>
> **[Ans]:** We apologize for causing a confusion. In Table 4, two editions of DAC-DETR ("w/ one-to-many Hungarian" and ours) actually use the same matching score definition (Eqn. 4). Their only difference is how to assign positive labels: the former uses set-to-set Hungarian while the latter (ours) uses the "threshold + ranking" criterion. Therefore, our superiority is not because better matching score, but is because C-Decoder favors the threshold strategy (L265 in the manuscript). We will clarify this point in the manuscript.

---

### Official Review · Reviewer_LpXx · 2023-07-06

**Soundness:** 4 excellent
**Presentation:** 3 good
**Contribution:** 3 good
**Rating:** 7
**Confidence:** 4

**Summary:**

This paper proposes a simple modification to the DETR architecture that improves upon several prior implementations. The paper identifies that the single decoder approach causes the model to try and achieve opposing objectives in terms of the query coverage and deduplication, and proposes a solution to the problem : an additional decoder having only cross attention layers is employed which improves performance on a standard benchmark (COCO).

**Strengths:**

## Originality and Significance
The findings about the self attention performing deduplication and cross attention gathering information from the image were already knows (as mentioned in this paper). The primary contribution here are
* Identifying that these two effects actually are in competition with each other, causing the training speed and final performance to be affected.
* Proposing a solution by adding an additional decoder having only cross attention.
* Displaying that this approach can be combined with several DETR variants and improves performance while reducing training time in all those cases
* Ablations showing that freezing weights in the O-decoder, adding self attention layers in C-decoder and removing the one-to-many matching all reduce performance, supporting the findings of the paper.
Adding an additional cross-attention only decoder is a simple modification that can be implemented on top of most other DETR variants, making the finding useful and significant. Since the added layers share weights with the original decoder, the size of the model does not increase, and the additional decoder is not used during inference, ensuring that there is no latency in inference time due to the proposed modification.

## Clarity and Quality
* Clear figures with descriptive stand-alone captions
* The findings are clearly explained with supporting evidence

**Weaknesses:**

* No analysis on if there is any difference in performance on large/small objects and common/rare objects
* Some qualitative examples of difference between the proposed approach and the baselines would be helpful

**Questions:**

1. What happens if you try to use the predictions from the C-decoder during inference as well? For instance by combining the predictions from the two decoders? Does it further improve performance?
2. Line 172 "and the second one is to suppress the label imbalance regarding different objects." Could you explain more clearly how this is achieved?
3. In Figure 3, is this average number of queries computed on the C-decoder or the O-decoder?


Suggestions:
1. Add an explanation of what t is in Figure 3 caption and change the y axis label to "avg number of queries / object"
2. The scale of the figures in Figure 3 should be kept constant, otherwise at first glance it looks like the gap increases at higher threshold, which is not the case
3. Writing suggestions: In several places, the phrase "the contrary" is used in a gramatically incorrect manner (divide the cross attention from the contrary / their contrary impairs ... etc) and I'd like to suggest that would be better substituted with the following phrasing:

"To improve the training efficacy, we propose a Divide-And-Conquer DETR (DAC-DETR) that divides the cross-attention out from this contrary for better conquering" --> "To improve the training efficacy, we propose a Divide-And-Conquer 6 DETR (DAC-DETR) that separates out the cross attention to avoid these competing objectives."

"have some contrary impacts on the object queries." -> "Have some opposing effects on the object queries"

"These two impacts are both critical for DETR (as explained later), and their contrary impairs the training efficacy." -> "These two impacts are both critical for DETR (as explained later), and their contrasting effects impairs..."

"In spite of the contrary, these two effects are both important for DETR." --> "In spite of the conflicting /rival objectives, we find that these two effects ...."

And similarly for other occurrences of the word contrary.

L169 grount-truth -> ground-thruth



**Limitations:**

Yes

---

> ### Author Rebuttal · Authors · 2023-08-10
>
> **LpXx-Q1: Analysis on if there is any difference in performance on large/small objects and common/rare objects.**
>
>  **[Ans]:**  Thanks for this suggestion. The performance gain on large/small objects is slightly larger/smaller (as shown in Table 1 in the manuscript), and the performance gain on common/rare objects is relatively smaller and larger (as shown in the table below). The details are as below:
>
> 1) Large objects vs small objects. Table 1 in the manuscript already provides detailed results on small ( AP$_S$ ), medium ( AP$_M$ ) and large ( AP$_L$ ) objects. It is observed that AP$_L$ has slightly larger improvement than ( AP$_S$ ). For example, on the Deformable-DETR baseline, the improvement on AP$_L$ and AP$_S$ is +4.3 AP and +3.0 AP, respectively. This is because small objects are inherently hard to detect and improving AP$_S$ is more difficult.
>
> 2) Common objects vs rare objects. We compare two most-frequent classes ('person' and 'car') against two rarest classes ('toaster' and 'hair direr') in COCO in the table below. It is observed that the improvement on the rare classes (+6.1 AP) is larger than on the common classes (+3.8 AP). We infer it is because the benefit of DAC-DETR (*i.e.*, improving the gathering effect) is more significant on rare classes.
>
>
>
> | Method | Common (person  \&  car)  |  Rare (toaster \&   hair direr)  |
> | :---: | :---: | :---: |
> | Baseline | 49.6 | 22.2 |
> | Variant-4  | 53.4 (+3.8) | 28.3 (+6.1) |
>
> * * *
>
> **LpXx-Q2: Some qualitative examples of difference between the proposed approach and the baselines would be helpful.**
>
> **[Ans]:** Thanks. Section A.6 (appendix) provides some qualitative examples of our DAC-DETR and the baseline. In the visualized examples, DAC-DETR shows better IoU and higher confidence on some hard examples (*e.g.*, occluded zebra, a cat wearing the hat). We think this is because DAC-DETR can improve both the quantity and quality of the queries for each object.
>
> * * *
>
> **LpXx-Q3: What happens if you try to use the predictions from the C-decoder during inference as well? For instance by combining the predictions from the two decoders? Does it further improve performance?**
>
> **[Ans]:** Thanks for this good suggestion. During rebuttal, we combine the predictions from the O-Decoder and C-Decoder and find it brings slight improvement (e.g., +0.3 AP). Specifically, we average the predicted logits of two decoders, use softmax to transform the averaged logits into classification scores and then use NMS to suppress the duplicate detections. On the Deformable-DETR baseline, while our DAC-DETR already achieves 47.1 AP (12 training epochs on CoCo), combining C-Decoder and O-Decoder brings another round of +0.3 AP improvement. We note that this improvement might become smaller on higher baselines and it considerably increases the inference cost.
>
>
> * * *
>
> **LpXx-Q4: Line 172 "and the second one is to suppress the label imbalance regarding different objects." Could you explain more clearly how this is achieved?**
>
> **[Ans]:** Thanks for your kind reminder. It is because an inherently easy-to-recognize object tends to attract many queries, therefore having many high-scored queries (which is also observed by DETA). Only using the threshold will make the easy-to-recognize objects have much more positive queries than the hard-to-recognize objects have, therefore causing label imbalance. Adding the top-$k$ selection suppresses the imbalance problem to some extent.
>
> * * *
>
> **LpXx-Q5: In Fig.3, is this average number of queries computed on the C-decoder or the O-decoder?**
>
> **[Ans]:** Thanks for the question. For our DAC-DETR, the average number of queries are computed on the C-decoder (no self-attention). As for the baseline, we remove its self-attention layers in the decoder on the already-trained model. It allows us to make a fair comparison on the gathering effect of two models. We will add these details to the manuscript.
>
> * * *
>
> **LpXx-Suggestions on figures and writings:**
>
> **[Ans]:** We sincerely thank your valuable suggestions and kind reminders. These suggestions help a lot in refining our manuscript. We will go through these details and make the revisions as below:
>
> 1) In Fig.3 , $t$ is the threshold on the matching score. We will add the explanation into the caption and change the y axis label to "avg number of queries / object".
>
> 2) We will make the scale of the figures constant in Fig. 3.
>
> 3) We appreciate that you pointed out the use of word "contrary" is actually inaccurate and grammatically incorrect. We will replace it with the suggested words (*e.g.*, conflicting objectives, opposing). We will carefully go through these typos and make our expression accurate.
>
> * * *

---

### Official Review · Reviewer_9psD · 2023-07-07

**Soundness:** 3 good
**Presentation:** 3 good
**Contribution:** 3 good
**Rating:** 7
**Confidence:** 4

**Summary:**

This paper reveals the “gather ↔ disperse” effects between cross-attention and self-attention layers in DETR decoder and proposes to add a decoder as the auxiliary branch without the self-attention blocks. The proposed approach achieves competitive detection performances.

**Strengths:**

1. this paper leverages the opposite effects of self-attention and cross-attention to build a auxiliary decoder and improves the detection performances

**Weaknesses:**

1. how general does the gather ↔ disperse effect apply? is there more examples beyond Figure 1
2. it's unclear if this can be combined with other practices such as stable matching and memory fusion to further improve the performances

**Questions:**

1. Is there any similarity between “gather ↔ disperse” phenomenon vs. the evolution algorithm with “prune out ↔ grow back”?
2. can this be combined with other practices such as stable matching and memory fusion to further improve the performances?
3. is it possible to add the C-decoder with self-attention module only in Table 3 for the completeness?

**Limitations:**

adequate

---

> ### Author Rebuttal · Authors · 2023-08-10
>
> **9psD-Q1: How general does the "gather$\leftrightarrow$disperse" effect apply? Is there more examples beyond Fig.1 ?**
>
>  **[Ans]:** Thanks for this good question. During rebuttal, we make a statistic across 20\% randomly-sampled images in COCO-2017 and validate that the "gather$\leftrightarrow$disperse" phenomenon is general. Specifically, we employ **1)** the averaged Euclidean distance between queries around the same object and **2)** the averaged IoU between each query and its nearby object, to measure the feature distance and positional distance, respectively. The results show that for the three cases (before the decoder, without decoder self-attention layers, and with decoder self-attention layers), the averaged feature distances are 22.58 $\rightarrow$ 12.35 (gather) $\rightarrow$ 14.13 (disperse) and the averaged IoU scores are 0.41 $\rightarrow$ 0.72 (gather) $\rightarrow$ 0.62 (disperse). These statistical results are consistent with our observation in Fig. 1 in the manuscript and show the "gather$\leftrightarrow$disperse" phenomenon is general. Moreover, we provide more visualizations in the supplementary PDF.
>
> * * *
>
> **9psD-Q2: It's unclear if this can be combined with other practices such as stable matching and memory fusion to further improve the performances**
>
>  **[Ans]:** Thank you for this suggestion. During rebuttal, we combine our DAC-DETR with the stable matching and memory fusion and observe clear (+0.7 AP) improvement. Specifically, DAC-DETR achieves 50.0 AP on the DINO baseline (COCO, 12 training epochs, ResNet-50 backbone). Adding the stable matching and memory fusion brings another round of +0.7 AP (50.0 $\rightarrow$ 50.7) improvement, showing good compatibility. We note that the Stable DINO's official code has not been released and the above experiment is based on our own implementation. Using its future official code might bring better results.
>
> * * *
>
> **9psD-Q3: Is there any similarity between "gather$\leftrightarrow$ disperse" phenomenon vs. the evolution algorithm  with "prune out$\leftrightarrow$grow back"?**
>
>  **[Ans]:** Thanks. Constructing analogy between the "gather$\leftrightarrow$disperse" phenomenon and the "prune out$\leftrightarrow$grow back" operation in evolution algorithm is insightful and interesting: the "disperse" is indeed similar to the "prune out". There is difference as well: in evolution algorithm, what have been pruned out are permanently removed and what grow back are actually new$^{[1]}$. In contrast, in DETR, the queries are not removed and will be gathered again, yielding conflict that compromises  the training efficiency.
>
> [1] Torsten Hoefler, et al. Sparsity in deep learning: Pruning and growth for efficient inference and training in neural networks, 2021.
>
> * * *
>
> **9psD-Q4: Is it possible to add the C-decoder with self-attention module only in Table 3 for the completeness?**
>
>  **[Ans]:**  Thanks. Following your suggestion, we build a variant whose C-Decoder has only self-attention layers (no cross-attention layers). As shown in the table below, this variant (Variant-4) is even lower than the baseline by -2.4 AP. It is reasonable because the cross-attention layer is prerequisite for the detector decoder to derive the object information. We will add the result into Table 3 (variants comparison) in the manuscript.
>
> | Method |Backbone  | epochs | AP | AP50 |
> | --- | :---: | :---: | :---: | :---: |
> | Baseline | R50 | 12 | 43.7 | 63.0 |
> | Variant-4  | R50 | 12 | 41.3 | 60.3 |
> | DAC-DETR (ours) | R50 | 12 | 47.1 | 64.8 |

---

### Author Rebuttal · Authors · 2023-08-10

**General response**

We thank all the reviewers for their valuable comments. We provide point-to-point responses to each reviewer, as well as a supplementary  PDF for some visualization results.

---

### Decision · Program_Chairs · 2023-09-21

**Decision:**

Accept (poster)

**Comment:**

The reviewers agree that the paper presents a valuable contribution. The main strengths of the paper are
- the idea that the effect of deduplication and gathering information are in conflict with each other, leading the authors to propose an additional decoder without self-attention.
- strong empirical performance.
While some reviewers noted that the approach adds some complexity to the model and the novelty of the architecture is limited, the value of the experiment outweigh these limitations and I recommend acceptance.